# Cohort profile: The Swedish Tattoo and Body Modifications Cohort (TABOO)

Christel Nielsen [1,2] Kristofer Andréasson,[3] H Olsson,[4] Malin Engfeldt,[1,5]
Anna Jöud [1,6]

June 30, 2021

¹Occupational and Environmental Medicine, Laboratory Medicine, Lund University, Lund, Sweden
²Clinical Pharmacology, Pharmacy and Environmental Medicine, Public Health, University of Southern Denmark, Odense, Denmark
³Rheumatology, Clinical Sciences, Lund University, Lund, Sweden
⁴Cancer Epidemiology, Clinical Sciences, Lund University, Lund, Sweden
⁵Occupational and Environmental Medicine, Region Skåne, Lund, Sweden
⁶Skåne University Hospital, Research and Education, Region Skåne, Lund, Sweden

**Correspondence to**
Dr Christel Nielsen;
christel.nielsen@med.lu.se

## ABSTRACT

**Purpose** The Swedish Tattoo and Body Modifications Cohort (TABOO) cohort was established to provide an infrastructure for epidemiological studies researching the role of tattoos and other body modifications as risk factors for adverse health outcomes. It is the first population-based cohort with detailed exposure assessment of decorative, cosmetic, and medical tattoos, piercing, scarification, henna tattoos, cosmetic laser treatments, hair dyeing, and sun habits. The level of detail in the exposure assessment of tattoos allows for investigation of crude dose–response relationships.

**Participants** The TABOO cohort includes 13 049 individuals that participated in a questionnaire survey conducted in 2021 (response rate 49%). Outcome data are retrieved from the National Patient Register, the National Prescribed Drug Register and the National Cause of Death Register. Participation in the registers is regulated by Swedish law, which eliminates the risk of loss to follow-up and associated selection bias.

**Findings to date** The tattoo prevalence in TABOO is 21%. The cohort is currently used to clarify the incidence of acute and long-lasting health complaints after tattooing based on self-reported data. Using register-based outcome data, we are investigating the role of tattoos as a risk factor for immune-mediated disease, including hypersensitisation, foreign body reactions and autoimmune conditions.

**Future plans** The register linkage will be renewed every third year to update the outcome data, and we have ethical approval to reapproach the responders with additional questionnaires.

## INTRODUCTION

Tattoos have become a mainstream phenomenon over the last decades. In the European Union, it is estimated that 12% of the general population, corresponding to 60 million individuals, is tattooed[1] and most people get their first tattoo at a young age.[2] The high prevalence of tattoos and the lifelong exposure to tattoo ink have brought safety concerns into the spotlight.

Tattoo inks are complex mixtures of chemicals that contain organic and inorganic colour pigments and their by-products, as well as solvents, preservatives and contaminants. Numerous reports of hazardous chemicals in tattoo inks have been published

### STRENGTHS AND LIMITATIONS OF THIS STUDY

⇒ The Swedish Tattoo and Body Modifications Cohort (TABOO) is the first population-based cohort specifically designed to study health effects of tattoos and other body modifications.
⇒ Exposure was assessed at baseline in 2021 using a structured questionnaire administered by Statistics Sweden (response rate 49%).
⇒ The cohort contains 13 049 participants and the tattoo prevalence is 21%.
⇒ The exposure assessment covers permanent tattoos (decorative, cosmetic, and medical), piercing, scarification, henna tattoos, cosmetic laser treatments, hair dyeing, and sun habits.
⇒ Outcome data are available from questionnaires and from National Authority Registers with full population coverage.

over the last years.[3–5] Coloured pigments in modern tattoo inks are dominated by azo or polycyclic compounds. This may be problematic, because some azo compounds are decomposed to toxic aromatic amines when exposed to solar radiation or laser treatment.[6–8] The major constituent of black ink is carbon black, accompanied by polycyclic aromatic hydrocarbons,[1] compounds that are classified as carcinogenic or potentially carcinogenic by the International Agency for Research on Cancer.[9 10]

During the tattooing process, large amounts of pigment are injected into the dermis. When the skin barrier is breached and foreign bodies intrude the body, the immune system is triggered. The immediate response involves a local reaction induced by immunological cells. As foreign bodies are phagocytised by immune cells, they are transported to local lymph nodes where a systemic immune response is initiated. Deposition of tattoo pigment in human lymph nodes has been confirmed,[11 12] and the liver, spleen and kidneys have been proposed as other plausible destinations.[13] Although research is limited, deposition of tattoo pigment in the liver has been established in mice.[14] Thus,

**Table 1** Sociodemographic characteristics of the 13 049 responders and the 13 702 non-responders at baseline in 2021

| | Responders | | Non-responders | |
|---|---|---|---|---|
| | n | (%)* | n | (%) |
| **Sex** | | | | |
| Male | 5719 | 45 | 7116 | 55 |
| Female | 7330 | 53 | 6586 | 47 |
| **Age** | | | | |
| 20–29 | 167 | 28 | 437 | 72 |
| 30–39 | 788 | 34 | 1510 | 66 |
| 40–49 | 1714 | 40 | 2622 | 61 |
| 50–59 | 4685 | 49 | 4859 | 51 |
| 60–69 | 5692 | 57 | 4271 | 43 |
| 70–79 | 3 | 50 | 3 | 50 |
| **Educational attainment** | | | | |
| Primary and lower secondary | 1061 | 32 | 2222 | 68 |
| Upper secondary | 5724 | 45 | 6916 | 55 |
| Postsecondary | 6243 | 60 | 4348 | 41 |
| Missing | 21 | 9 | 216 | 91 |
| **Country of birth** | | | | |
| Sweden | 11 393 | 53 | 9951 | 47 |
| Other | 1656 | 31 | 3751 | 69 |
| **Citizenship** | | | | |
| Swedish | 12 711 | 50 | 12 699 | 50 |
| Dual | 338 | 25 | 1003 | 75 |
| **Marital status** | | | | |
| Married | 7317 | 55 | 6051 | 45 |
| Unmarried | 3576 | 43 | 4828 | 57 |
| Divorced | 1941 | 43 | 2586 | 57 |
| Widowed | 215 | 48 | 237 | 52 |
| **Disposable income (SEK)** | | | | |
| None (0) | 248 | 25 | 743 | 75 |
| 1–124 999 | 586 | 30 | 1342 | 70 |
| 125 000–199 999 | 802 | 39 | 1243 | 61 |
| 200 000–279 999 | 1385 | 44 | 1793 | 56 |
| 280 000–369 999 | 2866 | 47 | 3221 | 53 |
| ≥370 000 | 7162 | 57 | 5360 | 43 |

*Percentages not summing to 100 are caused by rounding.

pigment exposure is not isolated to the tattoo site, but is systemic and may continue over months or even years as pigments decompose.[13]

Despite the high prevalence of tattoos, an improved understanding of the chemical composition of pigments, and an urgent need to understand if and how tattoos affect public health, epidemiological studies of adverse health outcomes are lacking. A limiting factor has been the absence of a sufficiently large cohort with adequate exposure assessment.

We performed three case–control studies, leveraging national Swedish registers and a questionnaire developed specifically for exposure assessment of tattoos, to investigate potential associations between tattoos and malignant melanoma, cutaneous squamous-cell carcinoma and lymphoma. The survey was performed in 2021. The Swedish Tattoo and Body Modifications Cohort (TABOO) was established by pooling the controls from these studies. It constitutes the largest population-based cohort with detailed exposure assessment of tattoos and other body modifications.

**Table 2** Data collection process and in-flow of responses over time

| Dispatch | Content | Date | Responders (n) | Response rate (%) |
|---|---|---|---|---|
| 1 | Study information Log-in instruction | 2 February | 6530 | 24.4 |
| 2 | Log-in instruction | 16 February | 1907 | 7.1 |
| 3 | Log-in instruction Paper questionnaire | 2 March | 3488 | 13.0 |
| 4 | Log-in instruction | 16 March | 1124 | 4.2 |
| Total | | 2 February–16 April | 13049 | 48.8 |

## COHORT DESCRIPTION

### Setting

The TABOO cohort uses a self-administered questionnaire for exposure assessment and leverages administrative Health Care Registers for outcome data. Information on potential confounders is available from the questionnaire and from registers.

Participation in Swedish Health Care Registers is regulated by the Act on Health Data Registers (1998:543) and several regulations (ie, 2001:707; 2001:709; 2005:363). Thus, the registers have full population coverage and there is no risk of selection bias. Furthermore, there is no loss to follow-up unless individuals emigrate from Sweden. Data sources are merged based on personal identity numbers, which are unique and permanent throughout life.[15]

### Participants

In the original case–control studies, 3000 primary cases aged 20–60 years at the date of diagnosis were identified for each cancer type by the National Board of Health and Welfare from the National Cancer Register in 2020. The index date ranged from 2007 to 2017, depending on the incidence of the specific neoplasm. Statistics Sweden identified three random, but sex-matched, controls per case from the Total Population Register using incidence-density sampling with age as the underlying time scale. Individuals with other malignancies than malignant melanoma, cutaneous squamous-cell carcinoma and lymphoma were eligible as controls. The study population was cross-referenced with the National Patient Register to exclude vulnerable individuals according to prespecified psychiatric diagnostic codes (ie, F20–29 in the International Statistical Classification of Diseases and Related Health Problems, 10th revision).

In total, 26751 controls were invited to participate and the overall response rate among controls was 49%, implying a final cohort size after pooling of 13049 participants. Baseline characteristics of responders and non-responders are presented in table 1. Responders were generally older, married individuals born in Sweden with higher educational attainment and income compared with non-responders. There was a slight overweight towards female participants.

### Exposure assessment

Self-reported exposure status was assessed through a questionnaire, administered by Statistics Sweden, between February and April 2021. The questionnaire was titled 'The role of new life-style factors in skin cancer, lymphoma and other diseases' to avoid bias from self-selection related to exposure status.

The questionnaire could be answered either on the web or on paper. The initial contact with the study population was through regular mail, where the participants received information about the study and log-in instructions to the digital questionnaire. Participants were informed that they consented to participation if they answered the questionnaire. Three postal reminders were made. The timeline of the data collection process is described in table 2.

The questionnaire comprised 143 questions. Statistics Sweden performed a metrological assessment of the questionnaire's disposition, question formulation and response options, to identify potential problems, analyse their consequences and provide solutions. After revision, the questionnaire was piloted on 28 tattooed individuals, including a tattoo artist, for content and clarity.

The questionnaire asked about tattoo status and characteristics, scarification, piercing, hair dyeing, henna tattoos and cosmetic laser treatments (including, but not limited to, tattoo removal). It addressed sun habits according to validated questions previously used in the Melanoma Inquiry of Southern Sweden cohort,[16] covering recreational and occupational sun habits, history of sunburn, use of tanning beds and use of sunscreen. We also asked whether participants' sun habits had changed after getting tattooed. Finally, we asked about skin characteristics, that is, pigmentation and reaction to sun exposure, and life-style factors such as smoking, snuff use and alcohol consumption. An overview of the variables in TABOO is given in box 1. For a full description of the variables, including response options and whom each question targeted, we refer to the online supplemental file 1.

The questionnaire defined tattoos as permanent body art obtained for decorative, cosmetic (ie, permanent, or semipermanent makeup, and microblading), or medical (eg, skin reconstruction after surgery) purpose. Respondents were explicitly asked to consider also removed tattoos. Exposure was assessed as a dichotomous variable, that is, presence or absence of at least one tattoo. To allow

**Box 1** Exposure data and personal characteristics in the Swedish Tattoo and Body Modifications Cohort (TABOO)

Tattoos
Tattoo status
Type (decorative, cosmetic, medical)
Age at first tattoo
Age at last tattoo
Anatomical site
Area of tattooed body surface
Experience level of tattooist
Number of tattoos
Number of sessions
Colour(s)
Geographical region
Tattooed under the influence of alcohol or drugs
Other body modifications
Piercing status
Scarification status
Henna tattoo status
Cosmetic laser treatment*

Anatomical site
Where treatment was performed
Number of treatments
Hair dyeing
Frequency, last 5 years
Age at first hair dyeing
Skin characteristics
Childhood eczema
Natural eye colour
Natural skin tone**
Freckles after sun exposure
Reaction to first sun exposure
Sun habits
Severe sunburn: <13 years; >13 years
Recreational sun travel: <13 years; >13 years
Sunscreen use: childhood; adult
Sunbed use: <13 years; >13 years
Occupational sun exposure (April–September)
Change of sun habits after being tattooed
Tobacco smoking and snuff use
Amount per day
Age when started
Age at quitting (if applicable)
Alcohol consumption

*For tattoo removal, hair removal, mole or hyperpigmentation removal, thread vein removal, facial wrinkles reduction, or other purposes. Specific questions included for each purpose.
**According to the Fitzpatrick Skin Phototype Classification.[17]

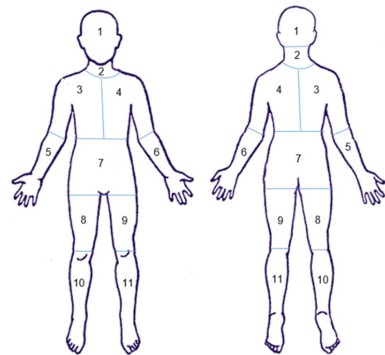

**Figure 1** Body manikin used to retrieve information on anatomical site and approximate area of tattooed body surface.

▶ Anatomical site of tattoos (according to a body manikin; figure 1).

The reliability of the exposure assessment was assessed in the pilot study. The pilot participants and a trained research assistant assessed the area of tattooed body surface at the same occasion. The weighted Cohen's kappa for the ordinal variable was estimated at 0.79 (95% CI 0.62 to 0.95), implying strong inter-rater agreement.

## Outcome data

The questionnaire addressed self-reported health complaints, that is, itch, redness, swelling, wounds and crusts, fever, papules, nodules, scaling, calluses, tenderness, pain, numbness, keloid scars and increased sensitivity to sunlight. We distinguished between health complaints that occurred during the healing process (ie, within 1 month after tattooing) and health complaints that developed with longer latency (ie, later than 1 month after tattooing). We asked whether the respondent sought care for each kind of complaint, and whether he or she turned to the tattooer or to the healthcare system. Similar, but less detailed, questions were asked with respect to piercing, scarification, and cause-specific laser treatments. Finally, we asked whether respondents searched for information on adverse health effects before getting tattooed, and whether the tattooist informed about any type of health risk (related to the healing process or long term) in connection with tattooing.

The health of the cohort is followed longitudinally in administrative population registers, including the National Patient Register, the National Drug Register and the National Cause of Death Register, all kept by the National Board of Health and Welfare. The registers have had national coverage since 1987, 2005 and 1961, respectively, and thus allow for both historical and prospective outcome assessment.

The first register linkage was performed in 2021. A specification of the retrieved outcome data is presented in table 3. The specific codes according to the International Classification of Diseases and the Anatomical Therapeutic Chemical underlying the data selection are given in the online supplemental file 1. Outcome data will

investigation of crude dose–response relationships, we assessed the area of tattooed body surface as:
▶ Increments on an ordinal scale (ie, less than 1 palm, 1–5 palms or more than 5 palms).
▶ Number of tattoos (≥20 cm to another tattoo).
▶ Number of tattoo sessions.

**Table 3** Characteristics of, and outcome data from, National Authority Registers linked to the Swedish Tattoo and Body Modifications Cohort (TABOO) in 2021

| | Content | Start year | Update frequency* | Coverage in 2020* | Data linked to TABOO | Outcomes | Study period |
|---|---|---|---|---|---|---|---|
| National Patient Register | Inpatient and specialised outpatient consultations to all healthcare professionals | Inpatient: 1987 Outpatient: 2001 | <2021: yearly>2021: monthly | Specific to each diagnostic code. Examples:Stroke 96%[18]Myocardial infarction 94%[19] | Events with specified diagnostic codes | Sarcoidosis Uveitis Secondary myocarditis Thyroiditis Rheumatoid arthritis Myositis Depression Anxiety Stress Abdominal pain Headache Back/neck pain Joint pain/myalgia Unspecified pain Persistent pain | Inpatient: 1990–2017 Outpatient: 2001–2017 |
| | | | | | Events within specific clinics | Dermatology and venereology Occupational dermatology | |
| National Prescribed Drug Register | Prescribed and dispensed drugs | 2005 | Monthly | Very high[20] | Events with specified drugs | Antibiotics Corticosteroids Dermatological agents Thyroid therapy Immunotherapy Immunoglobulin therapy Bile therapy Antiinflammatory and antirheumatic therapy Topical treatments for joint pain and myalgia Analgetics Neuroleptics Psychoanaleptics Adrenergics Antihistamines | 2005–2017 |
| National Cause of Death Register | Deaths in Sweden and among Swedish citizens abroad | 1961 | Yearly | Very high[21] | Death | Date | 2007–2017 |

*Compared with National Quality Registers.

be updated regularly through renewed register linkage every 3 years, and there is the possibility to expand the data frame by adding additional registers, variables and questionnaires.

TABOO has been linked to National Authority Registers to retrieve information on potential confounders. We have obtained yearly data on highest educational attainment, primary source of income, disposable income and marital status from the Longitudinal Integration Database for Health Insurance and Labour Market Studies (Statistics Sweden).

### Findings to date
The prevalence of tattoos in the cohort is 21% (n=2752). TABOO is currently used to clarify the incidences of acute and long-lasting health complaints after tattooing based on self-reported data. Using register-based outcome data, we are investigating the role of tattoos as a risk factor for immune-mediated disease, including hypersensitisation, foreign body reactions, and autoimmune diseases.

An updated list of publications based on TABOO will be published on the research group's web page: https://portal.research.lu.se/sv/organisations/epidemiology/publications/.

### Strengths and limitations
The main strength is that TABOO is the first population-based cohort with exposure data with respect to tattoos and other body modifications and a sufficient sample size, and hence statistical power, to investigate the role of these exposures as risk factors for a wide range of health outcomes, also those that are less common. The level of detail in the exposure assessment allows for investigation of crude dose–response relationships. Finally, the compulsory nature of the registers from

which outcome data is retrieved eliminates the risk of loss to follow-up and the potential selection bias that would follow.

As with any register-based study, the lack of biological samples is a limitation. However, observational studies are well suited to identify associations, generate more detailed hypotheses and pinpoint research questions that warrant further research.

**Acknowledgements** The authors are grateful to the pilot participants that provided constructive input to improve the clarity and validity of the questionnaire. We also express a big thank you to all the study participants that dedicated their time and effort to help us establish a unique research infrastructure that will move the field of health effects related to body modifications forward.

**Contributors** CN and AJ designed the study. CN, AJ, HO and ME constructed the questionnaire. CN directed the study's implementation. CN, AJ and KA defined the ICD and ATC codes that were retrieved from registers, and CN and AJ selected the sociodemographic variables. CN conducted the literature review and drafted the manuscript. All authors read and approved the final manuscript. CN is guarantor for the paper.

**Funding** This work was supported by the Swedish Research Council for Health, Working Life, and Welfare (2018-00864 to CN); the Craoford Foundation (20180659 to CN) and the Magnus Bergvall Foundation (201802669 to CN).

**Competing interests** None declared.

**Patient and public involvement** Patients and/or the public were involved in the design, or conduct, or reporting, or dissemination plans of this research. Refer to the Methods section for further details.

**Patient consent for publication** Not applicable.

**Ethics approval** This study involves human participants and was approved by Swedish Ethical Review Authority (no. 2019-03138). Participants gave informed consent to participate in the study before taking part.

**Provenance and peer review** Not commissioned; externally peer reviewed.

**Data availability statement** Data may be obtained from a third party and are not publicly available. The data underlying this article cannot be shared publicly due to the privacy of the participants. Register data are obtained from third parties and are not publicly available. However, data from the TABOO cohort can be made available for research collaboration upon reasonable request, and after approval from the Swedish Ethical Review Authority.

**ORCID iDs**
Christel Nielsen http://orcid.org/0000-0003-3940-7847
Anna Jöud http://orcid.org/0000-0001-7192-4911

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
