## [Reviewer comments · BMJ Open]

ARTICLE DETAILS

TITLE (PROVISIONAL)	Cohort Profile: The Swedish Tattoo and Body Modifications Cohort (TABOO)
AUTHORS	Nielsen, Christel; Andréasson, Kristofer; Olsson, H; Engfeldt, Malin; Jöud, Anna

VERSION 1 – REVIEW

REVIEWER	Foerster, Milena International Agency for Research on Cancer
REVIEW RETURNED	04-Jan-2023

GENERAL COMMENTS	Reviewer comments Major points: As you state, prospective cohorts on tattoo exposure and health outcomes are highly demanded and first of all, their approach to leverage their previously applied case-control design is highly appreciated. However, it is impossible to judge the data/study quality from the manuscript as it lacks crucial information, most notably the number of exposed individuals for different exposure scenarios as well as a power calculation for their major outcomes. From a first glance the sample size seems rather small, and it is unclear how it relates to the purpose of the study. Just as an example, we recently made a power calculation for our own studies on tattoos and long-term health effects, that are nested in the French and German national cohorts. Even in this far larger population (30,000 exposed, 90,000 non-exposed controls for a pooled analyses), the statistical power is only sufficient to detect a RR of 1.43 for NHL (annual incidence 22/100000, similarly rare like rheumatoid arthritis) after 10 years of follow-up. Without reporting both, exposure distributions and power calculations the manuscript cannot be interpreted with sufficient confidence, and accordingly major revisions are needed. Another major point concerns the exposure assessment. Although it is more detailed than in the few previous studies reporting on tattoo exposure, your statement that the collected data would allow for dose-response relationships seems unlikely. Apart from the presumably very low number of high exposed, it remains unclear how the tattooed body surface should be estimated from the collected data. Number of tattoos do not correlate well with tattoo surface and the three categories to measure tattoo surface in hand palms do not capture the full range of exposure. Moreover, the surface area cannot be linked to specific colours although tattoo ink toxicity might be colour dependent. Also, a very important factor to estimate tattooed body surface, the tattoo shading/filling, was not assessed. Before this background, more information is needed on how the authors intend to investigate dose-response relationships and how the levels of exposure may relate to the
---

“real” exposure. Maybe the questionnaire test measurements can be used to clarify this point?

Minor points

Please read carefully through the manuscript. I spotted some typos.

Please revise and extend your reference list. Some of the references are outdated and tattooing is a very dynamic topic. Also, in general no references explain the choice of outcomes.

Introduction:

- Metals, which are frequent contaminants (and sometimes still colour giving pigments) are not mentioned, neither is titanium dioxide.
- REACH is not mentioned
- Line 38-47: Although this seems to be a plausible scenario there is still not sufficient proof to present it as factual. Please rephrase in conjunctiva and include supporting literature for this hypothesis.
- There is no link to potential health effects made. Please include potential outcomes and why they might be of interest.

COHORT PROFILE:

- P 6, line 6: “ and an urgent need to understand how tattoos may affect public health” Please rephrase to “ and an urgent need to understand if and how tattoos could affect public health
- P 6, lines 13-20: “We performed three case-control studies, leveraging national Swedish registers and a questionnaire developed specifically for exposure assessment of tattoos, to investigate potential associations between tattoos and malignant melanoma, cutaneous squamous-cell carcinoma, and lymphoma. The survey was performed in 2021.” Do you already have the results of these three case control studies? They seem to be much more relevant than the relatively small TABOO cohort itself. If possible, include findings.

- P 6, lines 25-27: “It constitutes the largest population-based cohort with detailed exposure assessment of tattoos and other body modifications as well as adverse health outcomes.” This is not entirely true. As the authors might know, there are at least two large cohort studies on tattoos ongoing within the German and French national cohort (NAKO and Constances), both with far larger sample sizes and much more detailed exposure assessment. Initial tattoo information (yes/no) in both cohorts was already collected 2017-2020 and detailed exposure assessment in the tattooed is scheduled for mid-2023. In each of these cohorts approx. 15,000 participants are tattooed and the controls will be drawn from the non-tattooed population. Please revise accordingly and mention the national cohorts and their far larger sample size.

Participants:

- 37-42 : “Responders were generally older, married individuals born in Sweden with higher educational attainment and income.” – “Generally older” than non-responders (?) – please specify.

Besides, due to the design of the study , its population is not representative for the general population due to the very advanced age (controls of cancer cases). Please make this clear in your manuscript as you call it “population based”. This is even more important, as responders are older than non-responders (also marital status and education might be influential here). These population characteristics might lead to a very low tattoo and body modification prevalence in your cohort which you should, as mentioned in the beginning, describe in general.

- Page 12, line 11-18: “The reliability of the exposure assessment was assessed in the pilot study. The pilot participants and a

trained research assistant assessed the area of tattooed body surface at the same occasion. The weighted Cohen's kappa for the ordinal variable was estimated at 0.79 (95 % confidence interval: 0.62-0.95), implying strong inter-rater agreement." Please be more specific, how were these measurements done? And was their any objective measurement applied? (Measuring tattoos, image analyses etc)

For our cohort studies we developed a now validated questionnaire in collaboration with leading tattoo experts and epidemiologists, the Epidemiological Tattoo Assessment Tool (EpiTAT). Its development and validation will be published in a couple of days. In our validation study we found a considerable overestimation of self-reported tattoo size compared to objective validation measures and digital image analyses. Did you look at something like that in your test measurements?

Foerster, M., Dufour, L., Bäuml, W., Schreiver, I., Goldberg, M., Zins, M., Ezzedine, K., Schüz, J. "Development and Validation of the Epidemiological Tattoo Assessment Tool to Assess Ink Exposure and Related Factors in Tattooed Populations for Medical Research: Cross-sectional Validation Study" JMIR Formative Research, in press

- Page 11, line 39: "Number of tattoos (≥ 20 cm to another tattoo)". This seems like a very arbitrary measure. What was the rationale to include this? (Both for the number of tattoos and the >20 cm distance)

- Sufficient sample size: Please see my major revision recommendations. From a first glance the sample size seems quite small to address your research questions. Please include the exposure distribution and power calculation for at least some outcomes.

Exposure assessment:

P 11, lines 29-44 and table 2: "To allow investigations of dose-response relationships, we assessed the area of tattooed body surface as:

- Increments on ordinal scale (i.e., less than 1 palm, 1 to 5 palms, or more than 5 palms),
- Number of tattoos (≥ 20 cm to another tattoo),
- Number of tattoo sessions
- Anatomical site of tattoos (according to a body manikin..."

Given the complexity of tattoo exposure not sure whether these are sufficient to calculate meaningful dose-response relationships as it is proposed in the manuscript. Amongst others, tattoo exposure is influenced by tattoo size, shading/filling and colour, UV-exposure, artists expertise, and age of the tattoo. Very simplified, the total dose per person for tattooing exposure relates to the tattooed body surface, that could be estimated via the tattoo size (e.g. in hand palms) times the degree of tattoo filling. Unfortunately, the three "hand palm" categories are not sufficient to judge the total tattoo size and neither was the tattoo shading assessed. Not quite sure about the usefulness of the other variables: The number of tattoos only poorly correlates with tattoo size and the number of tattoo sessions might depend on many secondary factors (expertise of the artist, individual pain threshold, design of the tattoo etc).

Going through the exposure questionnaire, these are the main shortcomings that should be addressed in the limitations section.

- First, the tattoo size, most likely the most important factor of exposure, is assessed via only three exposure categories
- The tattoo colours cannot be linked to the size

	 • I cannot see shading/filling of tattoos in the questionnaire. As tattoo motives/types can vary from only outlines to completely filled there is a huge exposure variation coming with shading/filling. • The very common scenario of home tattooing/ tattoos done by lay persons is poorly assessed although it might be an exposure determining factor. • Laser treatment: not sure to understand why the questionnaire included different purposes of laser treatment ? While in regards to tattoos and body modifications these other purposes seem rather irrelevant, the important factor “removed tattoo colour” was not assessed although treatment toxicity depends on it. • Tattoo related infections (bacterial/viral/fungal) are not included... Outcome data: Please also provide the questionnaire items , e.g. as online material Table 4:  • Coverage should be reported for the specific outcomes instead of stroke and myocardial infections as they will not be studied • Generally the list of outcomes needs some explanation. Why were these chosen, what are the scientific hypotheses for the different types of outcomes (e.g. types of chronic pain or psychiatric outcomes)? Drugs: not sure how relevant this is in relation to your outcomes considering the very advanced age of your cohort. Please explain in the manuscript. Findings to date:  • P 29, line 11: “Primary outcomes are infection, inflammation, and persistent pain.” Except of the pain I could not figure out what exactly and how it is measured? Do you use the drug data? Please explain this in greater detail in the Outcomes section. Please be more specific here. Strengths and limitations: P 29, lines 34-41 “The main strength of TABOO is that it is the first population-based cohort with exposure data with respect to tattoos and other body modifications and sufficient size, and hence statistical power, to investigate the role of these exposures as risk factors for different health outcomes, also those that are less common.” Again, my major point: this needs to be proven in the manuscript via reporting on exposure distributions, power calculations and also how dose-response relationships should be assessed... Supplementary material: I recommend to spell out the ICD-10 diagnostic outcomes.
--	---

REVIEWER	Kluger, Nicolas HYKS sairaanhoitopiiri, Dermatology
REVIEW RETURNED	28-Feb-2023

GENERAL COMMENTS	Presentation of on going national study about tattooed individuals and various conditions. No specific comments. Regarding the reference and epidemiological data. A more recent article could add to the references. Otherwise no specific comment. Kluger N, Seité S, Taieb C. The prevalence of tattooing and motivations in five major countries over the world. J Eur Acad Dermatol Venereol. 2019 Dec;33(12):e484-e486.
--

VERSION 1 – AUTHOR RESPONSE

Reviewer 1

Major points:

As you state, prospective cohorts on tattoo exposure and health outcomes are highly demanded and first of all, their approach to leverage their previously applied case-control design is highly appreciated. However, it is impossible to judge the data/study quality from the manuscript as it lacks crucial information, most notably the number of exposed individuals for different exposure scenarios as well as a power calculation for their major outcomes.

From a first glance the sample size seems rather small, and it is unclear how it relates to the purpose of the study. Just as an example, we recently made a power calculation for our own studies on tattoos and long-term health effects, that are nested in the French and German national cohorts. Even in this far larger population (30,000 exposed, 90,000 non-exposed controls for a pooled analyses), the statistical power is only sufficient to detect a RR of 1.43 for NHL (annual incidence 22/100000, similarly rare like rheumatoid arthritis) after 10 years of follow-up. Without reporting both, exposure distributions and power calculations the manuscript cannot be interpreted with sufficient confidence, and accordingly major revisions are needed.

Another major point concerns the exposure assessment. Although it is more detailed than in the few previous studies reporting on tattoo exposure, your statement that the collected data would allow for dose-response relationships seems unlikely. Apart from the presumably very low number of high exposed, it remains unclear how the tattooed body surface should be estimated from the collected data. Number of tattoos do not correlate well with tattoo surface and the three categories to measure tattoo surface in hand palms do not capture the full range of exposure. Moreover, the surface area cannot be linked to specific colours although tattoo ink toxicity might be colour dependent. Also, a very important factor to estimate tattooed body surface, the tattoo shading/filling, was not assessed. Before this background, more information is needed on how the authors intend to investigate dose-response relationships and how the levels of exposure may relate to the “real” exposure. Maybe the questionnaire test measurements can be used to clarify this point?

AU:

- We are about to submit a paper on exposure characteristics and left the prevalences out of this manuscript to avoid double reporting. However, we acknowledge the reviewer’s point and have added the prevalence of tattoos under “Findings to date”.
- While power calculations have undisputed value in the design phase of inferential studies, they are irrelevant in a descriptive paper outlining the characteristics of an established cohort that can be used in future to investigate associations between various lifestyle factors and a multitude of different outcomes.
- The Swedish Authority Registers allow for prospective follow-up of the participants. In terms of person-years, the cohort is therefore very large and continuously expanding.
- TABOO allows for crude assessment of dose-response, i.e., tattooed body area can be addressed on the ordinal scale. We have clarified this by adding “crude” to the text.
- Epidemiological surveys must always balance the level of detail in exposure assessment against the risk of selective participation and item nonresponse. This trade-off is always a challenge that must be addressed a priori. We look forward towards increased understanding of this balance as the research field advances.

Minor points

Please read carefully through the manuscript. I spotted some typos.

Please revise and extend your reference list. Some of the references are outdated and tattooing is a very dynamic topic. Also, in general no references explain the choice of outcomes.

AU: We cite the key references that formed the rationale for the establishment of TABOO. Our intention was not to provide a comprehensive overview of the published literature, that has been done elsewhere.

The choice of the (initial) outcome variables presented in the manuscript was driven by the immune activation that occurs after skin barrier breach, as outlined in the Introduction. The cohort can be used to study basically any outcome in the future through renewed register linkage.

Introduction:

Metals, which are frequent contaminants (and sometimes still colour giving pigments) are not mentioned, neither is titanium dioxide.

AU: The Introduction states that tattoo ink contains both organic and inorganic colour pigments. We have added contaminants to the text.

REACH is not mentioned

AU: Correct. This is because we don't consider it relevant in the context of a cohort description. It should be stressed here that the TABOO data collection was performed before the implementation of the REACH-tattoo Annex in 2022. This will be an important point to raise in the discussions of future inferential studies.

Line 38-47: Although this seems to be a plausible scenario there is still not sufficient proof to present it as factual. Please rephrase in conjunctiva and include supporting literature for this hypothesis.

AU: The physiological mechanism of immune reactions to foreign bodies is generic knowledge, but we agree with the reviewer that it has yet to be demonstrated for tattoo ink. We have reformulated the section to reflect this.

There is no link to potential health effects made. Please include potential outcomes and why they might be of interest.

AU: The pathophysiological mechanisms are not known – which emphasizes the need for epidemiologic research. And again, TABOO is not limited to the study of a few outcomes as registry-linkages can be updated. We believe the case is made as we explain the content of human toxicants in tattoo ink, the systemic translocation of pigments, and the lack of epidemiologic studies.

COHORT PROFILE:

P 6, line 6: “ and an urgent need to understand how tattoos may affect public health” Please rephrase to “ and an urgent need to understand if and how tattoos could affect public health

AU: Changed.

P 6, lines 13-20: “We performed three case-control studies, leveraging national Swedish registers and

a questionnaire developed specifically for exposure assessment of tattoos, to investigate potential associations between tattoos and malignant melanoma, cutaneous squamous-cell carcinoma, and lymphoma. The survey was performed in 2021.” Do you already have the results of these three case control studies? They seem to be much more relevant than the relatively small TABOO cohort itself. If possible, include findings.

AU: The case-control studies are published separately. The TABOO is distinct from these studies as it is formed by pooling the controls from the three studies – it is not the same dataset.

P 6, lines 25-27: “It constitutes the largest population-based cohort with detailed exposure assessment of tattoos and other body modifications as well as adverse health outcomes.” This is not entirely true. As the authors might know, there are at least two large cohort studies on tattoos ongoing within the German and French national cohort (NAKO and Constances), both with far larger sample sizes and much more detailed exposure assessment. Initial tattoo information (yes/no) in both cohorts was already collected 2017-2020 and detailed exposure assessment in the tattooed is scheduled for mid-2023. In each of these cohorts approx. 15,000 participants are tattooed and the controls will be drawn from the non-tattooed population. Please revise accordingly and mention the national cohorts and their far larger sample size.

AU: TABOO remains the largest population-based study designed to specifically study health effects of tattoos and other body modifications. While we acknowledge the excellence of generic epidemiologic cohorts, such as NAKO and Constances, they also have their limitations. Most importantly, they require motivated participants as they include both interviews and physical examinations, which may introduce external validity issues. In addition, participants are made aware of the purpose of the study from exposure assessment, and there is an inherent risk of recall bias when using self-reported outcome data. In Sweden, we have the advantage of having administrative registers with full population coverage, which means that we can pull outcome data from health care visits that take place independently of the survey. Finally, we made sure not to condition participation on exposure by collecting data on multiple exposures related to new lifestyle factors.

That said, observational study designs have their pros and cons, and it is excellent that multiple designs are applied to answer this important research question. If we all reach the same conclusions, it would generate convincing evidence.

Participants:

37-42 : “Responders were generally older, married individuals born in Sweden with higher educational attainment and income.” – “Generally older” than non-responders (?) – please specify.

AU: Yes, this refers to the drop-out analysis. It has been clarified.

Besides, due to the design of the study , its population is not representative for the general population due to the very advanced age (controls of cancer cases). Please make this clear in your manuscript as you call it “population based”. This is even more important, as responders are older than non-responders (also marital status and education might be influential here). These population characteristics might lead to a very low tattoo and body modification prevalence in your cohort which you should, as mentioned in the beginning, describe in general.

AU: The selection of cancer cases was restricted to individuals aged 20-60 at disease onset, we did not include individuals of” very advanced age”. The age range of the TABOO participants was 20-79 years at the time of survey, with the majority being middle aged (see Table 1). The participants were sampled from the Total Population Register, which constitutes the total population at risk. The

prevalence of tattoos has been added to the text.

Page 12, line 11-18: "The reliability of the exposure assessment was assessed in the pilot study. The pilot participants and a trained research assistant assessed the area of tattooed body surface at the same occasion. The weighted Cohen's kappa for the ordinal variable was estimated at 0.79 (95 % confidence interval: 0.62-0.95), implying strong inter-rater agreement." Please be more specific, how were these measurements done? And was their any objective measurement applied? (Measuring tattoos, image analyses etc)

AU: Unfortunately, we do not understand the question. As the questionnaire assessed self-reported area of tattooed body surface, the point was to quantify the inter-observer agreement. This was done by having the participants and the trained assistant independently assess tattoo size according to the ordinal variable, and the kappa coefficient is the agreement between these two assessments.

For our cohort studies we developed a now validated questionnaire in collaboration with leading tattoo experts and epidemiologists, the Epidemiological Tattoo Assessment Tool (EpiTAT). Its development and validation will be published in a couple of days. In our validation study we found a considerable overestimation of self-reported tattoo size compared to objective validation measures and digital image analyses. Did you look at something like that in your test measurements?

Foerster, M., Dufour, L., Bäuml, W., Schreiber, I., Goldberg, M., Zins, M., Ezzedine, K., Schüz, J. "Development and Validation of the Epidemiological Tattoo Assessment Tool to Assess Ink Exposure and Related Factors in Tattooed Populations for Medical Research: Cross-sectional Validation Study" JMIR Formative Research, in press

AU: No, we validated against the assessment of the trained assistant. We did not see a need for a more detailed exposure assessment than the hand-palm variable, considering that there is no previous knowledge whatsoever, and that hand-palm rating is standard procedure in dermatology. Although we initially intended to use image analysis for exposure assessment, we refrained from this approach partly because of the substantial inter-individual variation in pigment dose per cm² and subsequent exposure misclassification bias, and partly because of ethical concerns pertaining to taking and sharing pictures of naked skin.

Page 11, line 39: "Number of tattoos (≥ 20 cm to another tattoo)". This seems like a very arbitrary measure. What was the rationale to include this? (Both for the number of tattoos and the >20 cm distance)

AU: Yes, and this is why it is not our primary exposure variable but merely included to allow comparison of different variables for exposure assessment (methodological approach) and to enable comparison with the literature in future research.

Sufficient sample size: Please see my major revision recommendations. From a first glance the sample size seems quite small to address your research questions. Please include the exposure distribution and power calculation for at least some outcomes.

AU: Not applicable, see previous comments.

Exposure assessment:

P 11, lines 29-44 and table 2: "To allow investigations of dose-response relationships, we assessed the area of tattooed body surface as:

- Increments on ordinal scale (i.e., less than 1 palm, 1 to 5 palms, or more than 5 palms),
- Number of tattoos (≥ 20 cm to another tattoo),
- Number of tattoo sessions

- Anatomical site of tattoos (according to a body manikin...”

Given the complexity of tattoo exposure not sure whether these are sufficient to calculate meaningful dose-response relationships as it is proposed in the manuscript. Amongst others, tattoo exposure is influenced by tattoo size, shading/filling and colour, UV-exposure, artists expertise, and age of the tattoo. Very simplified, the total dose per person for tattooing exposure relates to the tattooed body surface, that could be estimated via the tattoo size (e.g. in hand palms) times the degree of tattoo filling. Unfortunately, the three “hand palm” categories are not sufficient to judge the total tattoo size and neither was the tattoo shading assessed. Not quite sure about the usefulness of the other variables: The number of tattoos only poorly correlates with tattoo size and the number of tattoo sessions might depend on many secondary factors (expertise of the artist, individual pain threshold, design of the tattoo etc).

AU: We disagree with the reviewer, the exposure variables will allow for a crude assessment of dose-response (small, intermediate, large). Given that there is a complete lack of epidemiologic research today, TABOO cannot only be used to study dichotomized exposure, but can also explore whether size matter. If there is evidence of dose-dependent associations, those should be investigated further with more detailed (preferably measured) exposure in a dataset suited for that purpose.

Going through the exposure questionnaire, these are the main shortcomings that should be addressed in the limitations section.

- First, the tattoo size, most likely the most important factor of exposure, is assessed via only three exposure categories
- The tattoo colours cannot be linked to the size
- I cannot see shading/filling of tattoos in the questionnaire. As tattoo motives/types can vary from only outlines to completely filled there is a huge exposure variation coming with shading/filling.

AU: Correct, these are some of the compromises we made to avoid survey- or item- dropout. After piloting the questionnaires and discussions with tattoo artists, we made the judgement that this is the amount of detail that is feasible to get from a self-administered questionnaire.

- The very common scenario of home tattooing/ tattoos done by lay persons is poorly assessed although it might be an exposure determining factor.

AU: Skill level of tattooer and location for tattooing is included.

- Laser treatment: not sure to understand why the questionnaire included different purposes of laser treatment ? While in regards to tattoos and body modifications these other purposes seem rather irrelevant, the important factor “removed tattoo colour” was not assessed although treatment toxicity depends on it.

AU: TABOO is used by other researchers to study cosmetic laser complications. The questionnaire explicitly asks participants to also consider removed tattoos and asks about complications of laser removal.

- Tattoo related infections (bacterial/viral/fungal) are not included...

AU: It is covered both by self-reported symptoms indicative of infection among the health outcomes and by ICD-codes from the National Patient Register (NPR).

Outcome data: Please also provide the questionnaire items , e.g. as online material Table 4:

AU: It is not part of the cohort profile but will be published as a separate research article.

- Coverage should be reported for the specific outcomes instead of stroke and myocardial infections as they will not be studied

AU: NPR coverage has not been validated for all outcomes. We choose stroke and myocardial infection as there is published reports to refer the interested reader to.

- Generally the list of outcomes needs some explanation. Why were these chosen, what are the scientific hypotheses for the different types of outcomes (e.g. types of chronic pain or psychiatric outcomes)? Drugs: not sure how relevant this is in relation to your outcomes considering the very advanced age of your cohort. Please explain in the manuscript.

AU: We will leave the discussion on outcomes as that has been addressed in detail by now. Drugs are used together with diagnostic codes to define outcomes with specific healthcare trajectories. For instance, conditions treated in primary care are not registered in NPD but can be identified when continuous pharmaceutical treatment is necessary. Another example is administration of antibiotics to identify infections.

Findings to date:

- P 29, line 11: "Primary outcomes are infection, inflammation, and persistent pain." Except of the pain I could not figure out what exactly and how it is measured? Do you use the drug data? Please explain this in greater detail in the Outcomes section. Please be more specific here.

AU: These outcomes will be investigated in separate studies leveraging the TABOO cohort. In the context of the cohort profile paper, this is just a summary of ongoing research in the cohort.

Strengths and limitations:

P 29, lines 34-41 "The main strength of TABOO is that it is the first population-based cohort with exposure data with respect to tattoos and other body modifications and sufficient size, and hence statistical power, to investigate the role of these exposures as risk factors for different health outcomes, also those that are less common."

Again, my major point: this needs to be proven in the manuscript via reporting on exposure distributions, power calculations and also how dose-response relationships should be assessed...

AU: We have included exposure prevalences and clarified that crude dose-response relationships can be studied in TABOO.

Supplementary material: I recommend to spell out the ICD-10 diagnostic outcomes.

AU: TABOO is a dynamic cohort in that the outcome variables can be updated and revised at any time. We therefore found it unnecessary to spell out all the codes but would be happy to do so if requested. We leave this decision to the editor.